# AutoBalance: An Automatic Balancing Framework for Training Physics-Informed Neural Networks

## Abstract

Physics-Informed Neural Networks (PINNs) provide a powerful and general framework for solving Partial Differential Equations (PDEs) by embedding physical laws into loss functions. However, training PINNs is notoriously difficult due to the need to balance multiple loss terms, such as PDE residuals and boundary conditions, which often have conflicting objectives and vastly different curvatures. Existing methods address this issue by manipulating gradients before optimization (a "pre-combine" strategy). We argue that this approach is fundamentally limited, as forcing a single optimizer to process gradients from spectrally heterogeneous loss landscapes disrupts its internal preconditioning. In this work, we introduce AutoBalance, a novel "post-combine" training paradigm. AutoBalance assigns an independent adaptive optimizer to each loss component and aggregates the resulting preconditioned updates afterwards. Extensive experiments on challenging PDE benchmarks show that AutoBalance consistently outperforms existing frameworks, achieving significant reductions in solution error, as measured by both the MSE and $L^\infty$ norms. Moreover, AutoBalance is orthogonal to and complementary with other popular PINN methodologies, amplifying their effectiveness on demanding benchmarks.

## 1 Introduction

Physics-Informed Neural Networks (PINNs) (Raissi et al., 2019; Karniadakis et al., 2021) have emerged as a versatile framework for integrating physical laws with neural networks. By embedding governing Partial Differential Equations (PDEs) into loss functions via automatic differentiation, PINNs enable the solution of both forward and inverse problems without dense observational data. This paradigm combines two complementary strengths: the expressivity of neural networks for approximating complex solution spaces, and the loss function represented by PDE residuals together with boundary and initial conditions. Their effectiveness has been demonstrated across a wide range of domains, including heat transfer (Xu et al., 2023; Cai et al., 2021; Si & Yan, 2025b; Majumdar et al., 2025), solid mechanics (Hu et al., 2024; Faroughi et al., 2024), stochastic systems (Zhang et al., 2020; Chen et al., 2021), and uncertainty quantification (Yang & Perdikaris, 2019; Zhang et al., 2019; Yang et al., 2021).

PINNs provide a framework for solving PDEs by leveraging automatic differentiation of coordinate-based neural networks, or implicit neural representations (INRs), to compute derivatives. At the heart of this approach lies a PDE-driven loss formulation, in which the residuals of the governing equations, together with boundary and initial conditions, are combined into a single objective function to be minimized (Raissi et al., 2019; Karniadakis et al., 2021). Despite their conceptual elegance and broad adoption, training PINNs effectively remains a significant challenge. A central difficulty is the inherent imbalance among the competing loss components, which induces gradient pathologies during optimization (Wang et al., 2021; Liu et al., 2025). Such imbalance–arising from factors including numerical stiffness, heterogeneous loss landscapes, and the soft enforcement of PDE constraints–can cause dominant terms to drive parameter updates while critical physical laws remain under-enforced (Anagnostopoulos et al., 2024). Most existing remedies attempt to mitigate this issue by adaptively adjusting the weights of different loss terms (Wang et al., 2022; McClenny & Braga-Neto, 2023; Anagnostopoulos et al., 2024; Song et al., 2024; Si & Yan, 2025a). Nevertheless,

despite numerous proposed strategies, there is still no consensus on a universally effective method for balancing PINN training.

Multi-Task Learning (MTL) methods (Liu et al., 2019; 2021a; Yu et al., 2020; Liu et al., 2025) commonly aim to mitigate task conflicts by manipulating gradients from different losses. The prevailing strategy is to "pre-combine" these gradients—typically through re-weighting or gradient manipulation—and then apply a standard optimizer such as Adam. We argue that this "pre-combine" approach has an inherent limitation: it requires a single optimizer preconditioner to process gradients drawn from fundamentally different loss landscapes. In practice, the Hessian spectra of distinct loss components (e.g., interior versus boundary losses in PINNs) can vary drastically. Mixing such spectrally heterogeneous gradients produces a composite signal with scrambled curvature information, yielding an unstable or poorly calibrated preconditioner. As a result, a single second-moment estimate in Adam cannot adequately normalize updates across conflicting curvature profiles.

To address this limitation, we introduce an alternative "post-combine" paradigm, illustrated in the right panel. Instead of merging raw gradients beforehand, each task's loss is optimized independently with its own adaptive optimizer. This yields distinct, well-preconditioned update vectors, as each optimizer naturally adapts to the curvature of its respective loss landscape. The quadratic example in the figure highlights this advantage: when a well-conditioned loss component ($\mathcal{L}_1$) is coupled with an ill-conditioned one ($\mathcal{L}_2$), the conventional "pre-combine" strategy produces a slow, oscillatory trajectory on the composite landscape. In contrast, our "post-combine" 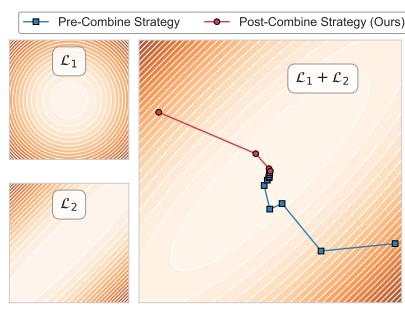 method moves efficiently toward the optimum. This ability to exploit curvature information more effectively forms the core idea that we formalize in Section 3.2.

Intriguingly, our key finding is that even the simplest aggregation—a direct summation of the individual updates—yields strong performance without the need for explicit balancing coefficients. We observe that this "decoupled" structure introduces an implicit balancing effect by naturally harmonizing the update norms across tasks. It is precisely this emergent, hyperparameter-free balancing property that underpins the robustness and effectiveness of our approach. We refer to this method as the *AutoBalance* Framework. Our contributions are summarized as follows:

- We identify a fundamental flaw in the prevailing "pre-combine" paradigm for PINN training: mixing gradients from loss components with heterogeneous Hessian spectra corrupts the preconditioner of adaptive optimizers. Building on this insight, we introduce a new "post-combine" framework, *AutoBalance*.

- We provide both analytical and empirical intuition of the effectiveness of our framework. Analytically, we show how AutoBalance accommodates heterogeneous Hessian spectra. Empirically, we demonstrate that AutoBalance stabilizes training by reducing the condition number of the preconditioned Hessian, yielding a smoother optimization landscape.

- We validate AutoBalance on a suite of challenging PDE benchmarks. Results show that AutoBalance provides a robust and versatile solution, consistently matching or outperforming the strongest baseline across a diverse suite of PDE benchmarks.

## 2 RELATED WORK

**Adaptive methods in PINNs.** In global weighting, Wang et al. (2021) introduced a learning-rate annealing strategy to dynamically adjust the weights of different loss terms, promoting balanced training. Building on gradient dynamics, Wang et al. (2022) employed Neural Tangent Kernel (NTK) analysis to guide weight adjustment, aligning learning with physical constraints. Xiang et al. (2022) applied Gaussian likelihood estimation to adaptively assign loss weights based on statistical properties, further enhancing weight allocation.

For point-wise weighting, McClenny & Braga-Neto (2023) introduced trainable point weights using a soft attention mask, where weights adapt to local loss via gradient descent/ascent. Song et al.

(2024) employed Latent Adversarial Networks (LANs) to learn adaptive point-error weights, improving PINN performance. Anagnostopoulos et al. (2024) proposed a gradient-free residual-based attention (RBA) scheme that updates weights directly from residual magnitudes with minimal overhead. Extending to temporal problems, Wang et al. (2024) developed Causality-PINN, assigning time-dependent weights for time-dependent PDEs. From an optimization view, Si & Yan (2025a) formulated point-wise weighting as a primal–dual method and implemented it through convolution.

**New architectures and new loss functions.** Beyond adaptive training strategies, researchers have proposed novel PINN architectures and refined loss formulations. For instance, Yu et al. (2020) penalized the gradient of the residual function, enforcing stronger adherence to the PDE. Wu et al. (2024) extended optimization from scattered points to their continuous neighborhoods via Monte Carlo sampling. Duan et al. (2025) dynamically reweighted PDE residual terms based on sample difficulty, measured by residual gradients. From an architectural perspective, Si et al. (2025) introduced a one-layer PINN inspired by the Cauchy activation function. Moreover, to better capture uncertainty, Wu et al. (2025) integrated fuzzy membership and fuzzy rule layers into PINNs.

**Balancing Strategies in Multi-Task Learning (MTL).** In multi-task learning (MTL), a central challenge is to identify a single, well-balanced Pareto-optimal solution. Existing approaches can be broadly grouped into two categories: loss balancing methods (Liu et al., 2019; Sener & Koltun, 2018; Liu et al., 2021b; Lin et al., 2023; Ye et al., 2021), which adjust task weights at the objective level, and gradient balancing methods (Sener & Koltun, 2018; Liu et al., 2021b; Yu et al., 2020; Liu et al., 2025; 2021a; Zhou et al., 2022; Fernando et al., 2023; Chen et al., 2018; Lin et al., 2023), which directly reconcile gradient magnitudes or directions to stabilize joint optimization.

Loss balancing methods dynamically adjust the relative weights of task losses during training. For instance, Dynamic Weight Average (DWA) (Liu et al., 2019) sets weights based on the rate of change of recent losses, assigning greater importance to tasks that are learning more slowly. Similarly, Impartial Multi-Task Learning (Liu et al., 2021b) normalizes loss magnitudes across tasks, preventing large-scale losses from dominating the optimization.

Gradient balancing methods instead act directly on task gradients, aiming to resolve conflicts and find a common descent direction. A seminal approach is the Multiple-Gradient Descent Algorithm (MGDA) (Sener & Koltun, 2018), which computes the minimal-norm vector within the convex hull of task gradients to avoid bias toward any single task. Extending this idea, IMTL-G (Liu et al., 2021b) seeks an update direction with equal projection magnitudes onto all task gradients. Projecting Conflicting Gradients (PCGrad) (Yu et al., 2020) identifies conflicting task pairs and removes the conflicting component by projecting one gradient onto the normal plane of the other. More recently, ConFIG (Liu et al., 2025) was introduced to address gradient conflicts in the specific setting of PINNs.

## 3 AUTOBALANCE: A "POST-COMBINE" STRATEGY FOR TRAINING PINN

### 3.1 THE CHALLENGE OF HETEROGENEOUS CURVATURES IN TRAINING PINN

PINNs are designed to solve systems of partial differential equations (PDEs) of the general form:

$$
\begin{aligned}
\mathcal{D}[u(x); x] &= 0, \quad x \in \Omega, \\
\mathcal{B}[u(x); x] &= 0, \quad x \in \partial\Omega,
\end{aligned}
\tag{1}
$$

where $u(x)$ denotes the unknown solution, $\mathcal{D}$ is the differential operator defining the PDE on the domain $\Omega \subseteq \mathbb{R}^d$, and $\mathcal{B}$ represents the boundary and initial conditions on $\partial\Omega$. The variable $x$ may also include time when considering time-dependent problems.

The core idea of PINNs is to approximate the unknown solution $u(x)$ with a neural network, typically a multilayer perceptron (MLP), denoted by $u(x; w)$, where $w \in \mathbb{R}^p$ are the network parameters. This reformulates solving a PDE as an optimization problem, in which the parameters $w$ are adjusted to minimize a composite loss function that enforces both the governing equations and the boundary conditions. The optimization problem is expressed as

$$
\underset{w \in \mathbb{R}^p}{\text{minimize}} \quad L(w) := \lambda_F L_{\text{res}}(w) + \lambda_B L_{\text{bc}}(w),
\tag{2}
$$

where the residual loss $L_{\text{res}}$ and the boundary loss $L_{\text{bc}}$ are evaluated at sets of collocation points $\{x_r^i\}_{i=1}^{N_f} \subset \Omega$ and $\{x_b^j\}_{j=1}^{N_b} \subset \partial\Omega$, respectively:

$$L_{\text{res}}(w) := \frac{1}{N_f} \sum_{i=1}^{N_f} \left( \mathcal{D}[u(x_r^i; w); x_r^i] \right)^2, \quad L_{\text{bc}}(w) := \frac{1}{N_b} \sum_{j=1}^{N_b} \left( \mathcal{B}[u(x_b^j; w); x_b^j] \right)^2. \quad (3)$$

The training objective of a PINN, as defined in Eq. (2), is the composite loss $L(w)$. Even in the absence of conflicts between the residual loss and the boundary loss—i.e., when a common minimizer $w^*$ exists for both $L_{\text{res}}$ and $L_{\text{bc}}$—convergence can still be hampered by differences in their respective loss landscapes. A major source of difficulty is the spectral heterogeneity of their Hessian matrices, $\nabla^2 L_{\text{res}}(w)$ and $\nabla^2 L_{\text{bc}}(w)$. In such scenarios, standard optimizers struggle to identify descent directions that ensure uniform progress. This challenge manifests as severe gradient imbalance during training: at the same point, one loss component may form steep "canyons" that generate excessively large gradients, while another may correspond to flat "plains" producing vanishingly small gradients. This disparity, arising from curvature mismatch, is a central factor underlying the difficulty of training PINNs.

To address the Hessian imbalance issue, we propose *AutoBalance*, a novel "post-combine" training paradigm. Instead of mixing raw gradients, AutoBalance first computes the gradient of each loss component using an independent adaptive optimizer. This enables each optimizer to construct a tailored preconditioner that matches the curvature of its corresponding loss landscape. The resulting update vectors are then aggregated. As an illustration, we provide an example of applying the AutoBalance framework to the AdamW (Loshchilov & Hutter, 2019) optimizer (see Algorithm 1 in the Appendix).

## 3.2 Curvature Balance: AutoBalance with Decoupled Curvatures

To understand why processing curvatures independently is beneficial from the curvature balance perspective, we analyze the optimization dynamics in a simplified setting of a quadratic objective. This simplification is well-motivated in the context of PINNs. Although the loss landscape of a PINN—typically an MLP with a Mean Squared Error (MSE) objective—is highly non-convex, prior research has shown that in the common overparameterized regime, all local minima are also global minima (Kawaguchi, 2016; Dauphin et al., 2014; Cooper, 2021; Achour et al., 2024). This justifies studying the complex optimization process by examining the local geometry around optimal solutions, which is well-approximated by a quadratic function characterized by the loss Hessian. Analyzing convergence on this quadratic model provides useful insights into the practical training dynamics of PINNs. To formalize our analysis, we consider a quadratic loss decomposed as

$$L(w) = L_1(w) + L_2(w); \quad L_1(w) := \frac{1}{2} w^T w, \quad L_2(w) := \frac{1}{2} w^T A^T A w, \quad (4)$$

where $A \in \mathbb{R}^{d \times d}$. In this construction, $L_2$ is essentially a scaled version of $L_1$, ensuring that both loss components share the same unique minimizer at the origin, $w^* = \mathbf{0}$.

This simplification allows us to isolate the impact of heterogeneous curvature (controlled by the matrix $A$) from the separate issue of conflicting minimizers. By focusing on this "pure" curvature mismatch scenario, we can analyze the optimizers' fundamental response to the landscape geometry. We defer a detailed discussion of conflict to Section 3.3. The total loss can thus be written in the compact form $L(\mathbf{w}) = \frac{1}{2} \mathbf{w}^T (I + A^T A) \mathbf{w}$, with $\tilde{H} = I + A^T A$ denoting the total Hessian of this composite loss. We now formalize the analysis to theoretically demonstrate the advantages of AutoBalance without bias correction (Algorithm 3) under scenarios with curvature imbalance, by focusing on the convergence rate for this simplified quadratic loss objective.

**Definition 1.** *An initialization $\mathbf{w}^0$ is said to satisfy* Bounded Initialization *for a quadratic function $f(\mathbf{w}) = \frac{1}{2} \mathbf{w}^T H \mathbf{w} + \mathbf{h}^T \mathbf{w}$ if the magnitudes of its initial gradient components are bounded relative to the Hessian spectrum $H$. Specifically, there exist constants $C_1, C_2 > 0$ such that for all components $i$:*

$$C_1 \lambda_{\max}(H) \leq |[\nabla f(w^0)]_i| \leq C_2 \lambda_{\max}(H).$$

**Theorem 1** (Convergence of Iterates in Euclidean Norm). *Consider AutoAdam without bias correction (Algorithm 3) and Adam without bias correction (Algorithm 2) applied to the quadratic loss*

$L(\boldsymbol{w})$ in equation 4, with $\beta_1 = 0$ and $\beta_2 = 1$. *Both algorithms produce iterates $\boldsymbol{w}^t$ that converge linearly to the minimizer $\boldsymbol{w}^* = \boldsymbol{0}$ in the Euclidean norm. By choosing an appropriate step size $\eta$, the one-step error reduction ratios for both methods satisfy the following bound:*

1. *For AutoAdam without bias correction, let $W = D_2^0(D_1^0)^{-1} + A^T A$. The convergence in the weighted norm $\|\cdot\|_W$ is bounded by*

$$\frac{\|\boldsymbol{w}^{t+1}\|_W}{\|\boldsymbol{w}^t\|_W} \leq \frac{\kappa((D_1^0)^{-1} + (D_2^0)^{-1/2} A^T A (D_2^0)^{-1/2}) - 1}{\kappa((D_1^0)^{-1} + (D_2^0)^{-1/2} A^T A (D_2^0)^{-1/2}) + 1},$$

*where $\kappa(\cdot)$ denotes the condition number of a matrix.*

2. *For Adam without bias correction, let $W_{Adam} = I + A^T A$. The convergence in the weighted norm $\|\cdot\|_{W_{Adam}}$ is bounded by*

$$\frac{\|\boldsymbol{w}^{t+1}\|_{W_{Adam}}}{\|\boldsymbol{w}^t\|_{W_{Adam}}} \leq \frac{\kappa((D^0)^{-1/2}(I + A^T A)(D^0)^{-1/2}) - 1}{\kappa((D^0)^{-1/2}(I + A^T A)(D^0)^{-1/2}) + 1}.$$

**Corollary 1.1.** *Suppose the AutoAdam initialization $\boldsymbol{w}^0$ is a Bounded Initialization for $L_1$ with constants $C_{1,1}, C_{1,2}$ and for $L_2$ with constants $C_{2,1}, C_{2,2}$. Then the one-step convergence ratio is upper bounded by*

$$\frac{\frac{C_{1,2}}{C_{1,1}} + \frac{C_{2,2}}{C_{2,1}} - 1}{\frac{C_{1,2}}{C_{1,1}} + \frac{C_{2,2}}{C_{2,1}} + 1}.$$

*Similarly, assume the AutoAdam initialization $\boldsymbol{w}^0$ is a Bounded Initialization for the combined loss $L$ with constants $C_1, C_2$. Then the convergence ratio is upper bounded by*

$$\frac{\frac{C_2}{C_1}\kappa(I + A^T A) - 1}{\frac{C_2}{C_1}\kappa(I + A^T A) + 1}.$$

The proofs for Theorem 1 and Corollary 1.1 are provided in Appendix B. The theory above offers a simple motivation for why AutoBalance performs well on problems with imbalanced curvatures. To illustrate the main idea, consider a special case where the problem has one well-conditioned component, $L_1$ (i.e., $\boldsymbol{w}^T \boldsymbol{w}$), and one ill-conditioned component, $L_2$ (induced by a matrix $A$ with $\kappa(A^T A) \gg 1$). In this scenario, Corollary 1.1 highlights a clear difference in convergence rates. The rate for standard Adam deteriorates due to the ill-conditioned component, as it depends on a large condition number associated with $\kappa(A^T A)$. In contrast, the convergence rate for AutoBalance depends primarily on the Bounded Initialization constants. This simplified result illustrates the core principle of our method: by employing separate preconditioners, AutoBalance leverages the good structure of the well-conditioned task without being adversely affected by the poorly conditioned component. This theoretical insight motivates the method's effectiveness in more complex, practical problems. To verify whether this scenario arises in PINN training, we empirically study the Hessian dynamics during the training of a PINN for the 2D Helmholtz equation. The results in Figure 1 confirm both the presence of this curvature imbalance and the effectiveness of our solution.

First, Figure 1 (left) validates our central premise. At initialization, the Hessian spectra of the residual loss ($L_{\text{res}}$) and the boundary loss ($L_{\text{bc}}$) are starkly different, confirming the spectral heterogeneity that motivates our work. In this case, the boundary loss exhibits a significantly wider eigenvalue distribution, indicative of poorer conditioning.

Next, we examine the impact of AutoBalance's "post-combine" preconditioning compared to standard Adam's "pre-combine" approach. The benefits are twofold. After training is complete, the spectrum of the AutoBalance-preconditioned Hessian is more favorably structured, with a larger minimum eigenvalue ($\lambda_{\min}$) than both the original and the standard Adam-preconditioned Hessians (Figure 1, center). This directly leads to a smaller condition number. This advantage holds throughout the entire training process. Figure 1 (right) tracks the evolution of the boundary loss condition number, showing that while ill-conditioning tends to worsen for all methods, AutoBalance consistently maintains a significantly lower effective condition number.

These empirical results provide strong evidence for our theoretical intuition. By designing a tailored preconditioner for each loss component, AutoBalance effectively mitigates the severe ill-conditioning that arises from heterogeneous curvatures, thereby creating a more stable and efficient optimization landscape for training PINNs.

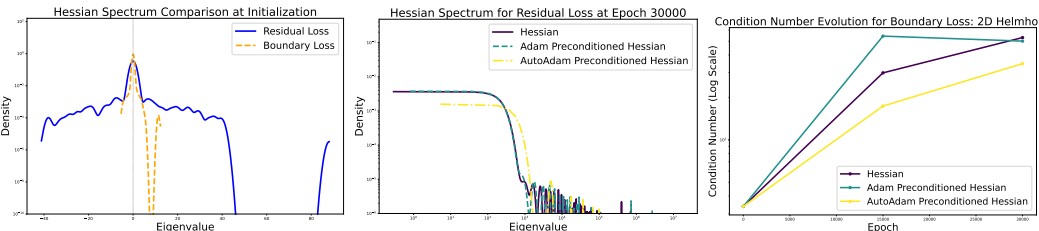

Figure 1: Empirical analysis of Hessian properties for the 2D Helmholtz equation. **(Left)** At initialization, the Hessian spectra for the residual and boundary losses exhibit strong heterogeneity. **(Middle)** After 30,000 epochs, the AutoBalance-preconditioned Hessian for the residual loss shows a larger minimum eigenvalue compared to both the original and Adam-preconditioned Hessians. **(Right)** During training, AutoBalance consistently maintains a substantially lower effective condition number for the boundary loss than standard Adam or the original Hessian.

### 3.3 UPDATE BALANCE: AUTOBALANCE WITH DIRECTION AND SCALE

Having addressed the challenge of heterogeneous curvatures, a natural question arises: how should the resulting updates from residual loss and boundary loss be balanced in terms of their respective magnitudes and directions? One might assume that an explicit balancing or gradient manipulation method would still be necessary to prevent one task from dominating the other.

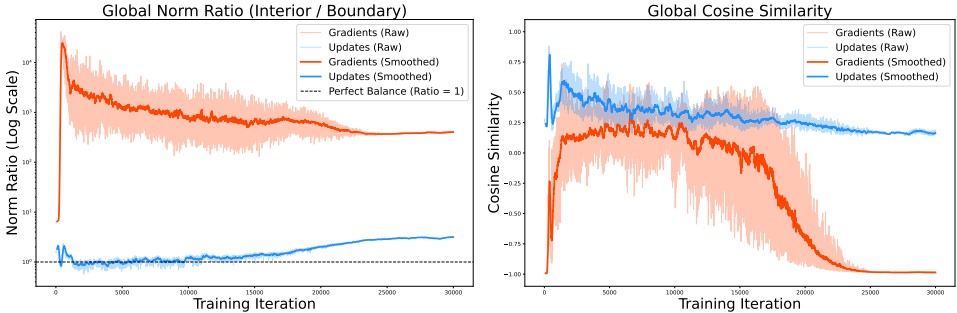

Figure 2: **AutoBalance exhibits an emergent auto-balancing property on the 2D Helmholtz problem. (Left)** The norm ratio of the raw gradients (interior/boundary) is highly imbalanced, whereas the ratio of the update vectors from AutoBalance consistently remains near the ideal balance of 1.0. **(Right)** The raw gradients become increasingly anti-aligned (negative cosine similarity), indicating task conflict. In contrast, the update vectors from AutoBalance maintain a positive alignment, ensuring constructive updates.

Intriguingly, we find that the AutoBalance paradigm exhibits an emergent auto-balancing property without any explicit intervention. We demonstrate this phenomenon empirically in Figure 2. For this analysis, we compute "global" gradient and update vectors for each loss component by concatenating the values for all network parameters at each training step. We then analyze two key metrics for a 2D Helmholtz problem: the ratio of these global vector norms and their cosine similarity.

Figure 2 (left) plots the norm ratio. The raw gradients exhibit a severe magnitude imbalance, with their norm ratio fluctuating by several orders of magnitude. This observation empirically confirms the well-known issue that motivates many loss-balancing techniques (Wang et al., 2022; Liu et al., 2019) in PINN training. In stark contrast, the norm ratio of the update vectors produced by AutoBalance remains remarkably close to the ideal value of 1.0. This shows that the effective contributions of the two tasks are automatically harmonized in magnitude.

Figure 2 (right) shows that this harmonizing effect extends to the update directions. The raw gradients become increasingly anti-aligned as training progresses, with their cosine similarity dropping towards -1. This increasing directional conflict suggests that methods focusing only on balancing

gradient magnitudes (like many loss-weighting schemes) may be insufficient to resolve the underlying task conflicts. The update vectors generated by AutoBalance, however, maintain a consistently positive cosine similarity, ensuring the updates for both tasks are constructive and do not work at cross-purposes.

This potent auto-balancing behavior is not an external mechanism but an intrinsic property of the adaptive optimizers themselves. The per-parameter normalization in Adam (i.e., division by the square root of the second-moment estimate, $\sqrt{\hat{v}_t} + \epsilon$) automatically scales the update. If a loss component consistently produces large or noisy gradients, its corresponding second-moment estimate will grow, effectively reducing its update magnitude. This dynamic process naturally balances the effective learning rates across all tasks, leading to a more stable and robust training trajectory without the need for any balancing hyperparameters.

## 4 EXPERIMENTS AND RESULTS

### 4.1 EXPERIMENTS SETTING

**Benchmarks.** For a comprehensive evaluation, we consider three benchmarks: 1D reaction-diffusion system, 2D high-frequency Helmholtz equation, and inverse problem for the 2D Poisson equation. (The formulation of the inverse problem differs slightly from that of PINNs for forward problems; details are provided in Appendix A.1).

**Baseline.** To demonstrate that our method effectively addresses the imbalance issue, we compare AutoBalance with several loss-balancing methods, including Equal Weighting (EW)[1], NTK-PINN (Wang et al., 2022), Dynamic Weight Average (DWA) (Liu et al., 2019), as well as gradient-balancing methods such as MGDA (Désidéri, 2012), PCGrad (Yu et al., 2020), IMTL-G (Liu et al., 2021b), and ConFIG (Liu et al., 2025). Also, since AutoBalance is directly applied to optimizers, we implement it on the AdamW optimizer within the AutoBalance framework (Algorithm 1).

Moreover, to evaluate the robustness of the AutoBalance method, we apply it to several PINN variants, including **adaptive weighting method** (RBA-PINN (Anagnostopoulos et al., 2024), CWP-PINN (Si & Yan, 2025a), and CoPINN (Duan et al., 2025)), **novel loss functions** (gPINN (Yu et al., 2020), RoPINN (Wu et al., 2024)), and a **new architecture** (DF-PINN (Wu et al., 2025)). Most PDE benchmarks we consider involve two losses: the residual loss and the boundary loss. However, gPINN introduces an additional term, the gradient of the residual loss, resulting in a three-loss setting. Our AutoBalance is designed to operate over a general number of losses $n$, so while many PDE cases involve only two, its application to gPINN (with three) highlights our method's generality.

### 4.2 MAIN RESULTS

**Comparable to balancing methods.** Table 1 and Fig. 3 summarize the performance of AutoBalance compared to existing loss-balancing and gradient-balancing baselines across three representative PDE benchmarks. For the 1D reaction-diffusion system, AutoBalance achieves the lowest MSE among all methods. The advantage becomes more pronounced on the more challenging 2D Helmholtz equation, where AutoBalance not only attains the smallest MSE but also maintains competitive $L^\infty$ error, demonstrating its ability to stabilize training for high-frequency solutions. Similarly, in the 2D Poisson inverse problem, AutoBalance consistently achieves the lowest MSE and $L^\infty$ error. Overall, these results highlight that AutoBalance is robust across diverse PDE settings and consistently enhances optimizer performance, providing a reliable and broadly applicable training improvement for PINNs.

To complement the aggregate results, we provide qualitative visualizations, such as heatmaps, in Appendix A.4.1 and A.4.2. Taking the 2D Poisson inverse problem as an example (Figure 6 in the Appendix), a clear performance hierarchy emerges. The reconstruction obtained with AutoAdamW (second row) faithfully recovers the structures in the ground truth, producing a visually accurate solution with the lowest point-wise error. In contrast, the baseline method DWA exhibits noticeably higher errors, while IMTL-G fails to reproduce the field's symmetry, yielding a distorted reconstruction with the highest error among the three methods.

---

[1]Equal Weighting (EW) chooses the same weight for all different loss functions.

Table 1: Comparison of Loss-Balancing and Gradient-Balancing Baselines

| PDE Category | Balancing Category | Baselines | AdamW | |
|---|---|---|---|---|
| | | | MSE | $L^\infty$ |
| **1D reaction-diffusion system** | **Loss Balance** | **EW** | 9.16e-7 | 1.15e-2 |
| | | **NTK**(Wang et al., 2022) | 6.31e-7 | 2.52e-3 |
| | | **DWA**(Liu et al., 2019) | 3.03e-7 | 5.51e-3 |
| | **Gradient Balance** | **PCGrad** (Yu et al., 2020) | 9.56e-6 | 1.65e-2 |
| | | **IMTL-G**(Liu et al., 2021b) | 2.32e-5 | 2.64e-2 |
| | | **ConFIG** (Liu et al., 2025) | 1.03e-6 | 5.52e-3 |
| | | **MGDA** (Désidéri, 2012) | 1.09e-5 | 9.56e-3 |
| | **Our Model** | **AutoBalance** | **2.61e-7** | 5.05e-3 |
| **2D Helmholtz equation** | **Loss Balance** | **EW** | 29.35 | 11.52 |
| | | **NTK** (Wang et al., 2022) | 6.94e-1 | 2.75 |
| | | **DWA**(Liu et al., 2019) | 12.74 | 12.28 |
| | **Gradient Balance** | **PCGrad**(Yu et al., 2020) | 5.05e-3 | 5.20e-1 |
| | | **IMTL-G**(Liu et al., 2021b) | 5.45e-3 | 4.84e-1 |
| | | **ConFIG** (Liu et al., 2025) | 2.23e-3 | 2.68e-1 |
| | | **MGDA**(Désidéri, 2012) | 8.17e-3 | 4.17e-1 |
| | **Our Model** | **AutoBalance** | **2.04e-3** | 3.84e-1 |
| **2D Poisson inverse problem** (diffusion coefficient $a(x, y)$) | **Loss Balance** | **EW** | 2.37e-3 | 1.07e-1 |
| | | **NTK** (Wang et al., 2022) | 2.71e-5 | 1.29e-2 |
| | | **DWA** (Liu et al., 2019) | 2.45e-5 | 1.27e-2 |
| | **Gradient Balance** | **PCGrad**(Yu et al., 2020) | 4.73e-5 | 1.64e-2 |
| | | **IMTL-G**(Liu et al., 2021b) | 7.34e-5 | 2.27e-2 |
| | | **ConFIG**(Liu et al., 2025) | 8.96e-6 | 8.40e-3 |
| | | **MGDA**(Désidéri, 2012) | 5.74e-5 | 1.82e-2 |
| | **Our Model** | **AutoBalance** | **6.19e-6** | 6.09e-3 |

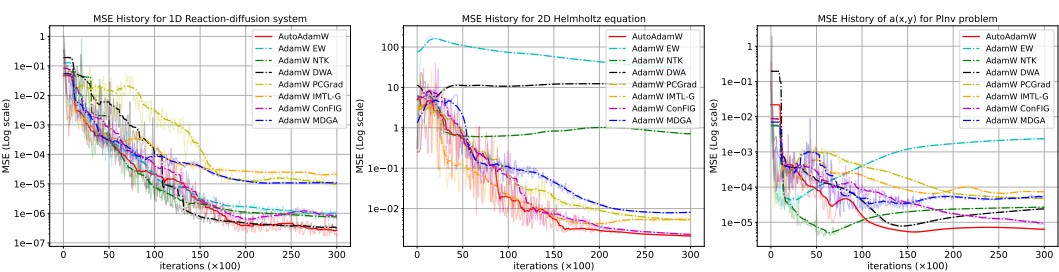

Figure 3: MSE history of AutoBalance and baseline methods for three PDE benchmarks: 1D reaction-diffusion system (Left), 2D Helmholtz equation (Middle), and 2D Poisson inverse problem (Right).

These findings further highlight that the optimal balancing strategy is problem-dependent. Loss-balancing methods perform well on the 1D reaction–diffusion system but degrade significantly on the 2D Helmholtz equation, where boundary constraints likely induce training instability. Gradient-balancing methods help mitigate this instability for Helmholtz but do not consistently retain their advantages across other settings, such as the Poisson inverse problem or the 1D reaction–diffusion system. AutoBalance's key strength lies in its ability to navigate these trade-offs automatically: it matches the top loss-balancing methods on diffusion problems while outperforming all gradient-balancing methods on Helmholtz, providing a robust, general-purpose solution.

**Orthogonal to PINN baseline models.** AutoBalance operates at the level of the optimizer and its preconditioning mechanism, making it directly applicable to various PINN variants. We evaluate its performance by comparing traditional AdamW with Auto-AdamW across several state-of-the-art PINN models developed in recent years (2020–2025). The numerical results are summarized in Table 2.

Table 2: Comparison of MSE and $L^\infty$ norms for AutoBalance versus state-of-the-art PINN baselines across multiple PDE benchmarks. Percentages in parentheses indicate the error relative to the baseline (current/baseline $\times$ 100); lower values indicate better performance.

| PDE Category | Training Method | MSE | | $L^\infty$ Norm | |
|---|---|---|---|---|---|
| | | w/o AutoBalance | with AutoBalance | w/o AutoBalance | with AutoBalance |
| **1D reaction-diffusion** | RBA-PINN(Anagnostopoulos et al., 2024) | 7.25e-6 | **2.82e-7** (3.9%) | 4.13e-3 | **3.71e-3** (89.8%) |
| | CWP-PINN(Si & Yan, 2025a) | 6.64e-6 | 9.26e-7 (14.0%) | 1.16e-2 | 8.76e-3 (75.5%) |
| | gPINN (Yu et al., 2020) | 1.38e-6 | 9.79e-7 (70.1%) | 1.17e-2 | 1.06e-2 (90.5%) |
| | RoPINN(Wu et al., 2024) | 1.04e-6 | 5.75e-7 (55.3%) | 6.64e-3 | 3.65e-3 (54.9%) |
| | CoPINN (Duan et al., 2025) | 1.92e-7 | 1.68e-7 (87.5%) | 1.60e-3 | 1.46e-3 (91.2%) |
| | DF-PINN (Wu et al., 2025) | 1.14e-6 | 5.06e-7 (44.4%) | 1.50e-2 | 3.94e-3 (26.3%) |
| **2D Helmholtz** ($a_1 = 5$, $a_2 = 5$) | RBA-PINN(Anagnostopoulos et al., 2024) | 3.49 | 1.25e-3 (0.04%) | 7.23 | 2.35e-1 (3.3%) |
| | CWP-PINN (Si & Yan, 2025a) | 6.27e-3 | **8.79e-4** (14.0%) | 6.18e-1 | **2.78e-1** (44.9%) |
| | gPINN (Yu et al., 2020) | 10.2 | 7.80e-3 (0.08%) | 33.4 | 3.23e-1 (0.97%) |
| | RoPINN (Wu et al., 2024) | 7.40e-3 | 1.69e-3 (22.8%) | 7.02e-1 | 4.24e-1 (60.4%) |
| | CoPINN (Duan et al., 2025) | 9.61e-3 | 1.31e-3 (13.6%) | 9.97e-1 | 3.43e-1 (34.4%) |
| | DF-PINN (Wu et al., 2025) | 5.63e-3 | 9.83e-4 (17.5%) | 4.29e-1 | 2.96e-1 (68.9%) |
| **2D Poissoon inverse Problem** (diffusion coefficient $a(x, y)$) | RBA-PINN(Anagnostopoulos et al., 2024) | 1.88e-5 | 3.48e-6 (18.5%) | 9.83e-3 | 4.03e-3 (40.9%) |
| | CWP-PINN (Si & Yan, 2025a) | 4.83e-6 | **1.19e-6** (24.6%) | 5.25e-3 | **2.21e-3** (42.1%) |
| | gPINN (Yu et al., 2020) | 5.51e-5 | 1.01e-5 (18.3%) | 2.33e-2 | 8.89e-3 (38.2%) |
| | RoPINN (Wu et al., 2024) | 1.34e-5 | 4.56e-6 (34.0%) | 1.05e-2 | 6.33e-3 (60.3%) |
| | CoPINN (Duan et al., 2025) | 4.32e-6 | 2.02e-6 (46.7%) | 5.65e-3 | 3.94e-3 (69.7%) |
| | DF-PINN (Wu et al., 2025) | 2.17e-5 | 7.84e-6 (36.1%) | 1.08e-2 | 7.43e-3 (68.8%) |

The results provide strong evidence that AutoBalance is fundamentally orthogonal to architectural innovations in PINNs. Regardless of whether a model employs point-wise weighting schemes (CWP-PINN, RBA-PINN, CoPINN), gradient-enhanced losses (gPINN), region-based enhancements (RoPINN), or novel architectures (DF-PINN), it still relies on an optimizer to navigate the loss landscape. AutoBalance directly improves this optimization process, leading to consistent performance gains ranging from moderate to substantial. Notably, on the challenging high-frequency 2D Helmholtz problem, AutoBalance significantly stabilizes training and enhances accuracy for all models, including those that fail without it. These results confirm that AutoBalance is not a competitor to existing state-of-the-art models but a complementary tool that can be applied on top to further boost their performance.

## 5 CONCLUSION

In this work, we addressed the optimization challenges in training Physics-Informed Neural Networks arising from composite losses with heterogeneous curvatures. To tackle this, we proposed AutoBalance, a novel "post-combine" training framework that decouples preconditioning for each loss component. We established the theoretical intuition for our approach by analyzing its convergence on a simplified curvature-imbalanced quadratic problem. We then empirically verified its ability to balance both curvature and gradients by observing the evolution of the Hessian spectrum, update norm ratios, and cosine similarities during training. Moreover, our empirical evaluation demonstrated the robustness and practical benefits of this approach: AutoBalance consistently outperformed various gradient-balancing methods across multiple challenging PINN variants. This work highlights the importance of moving beyond solely managing directional gradient conflicts, toward a strategy that adapts to the unique curvature of each loss landscape. While effective, AutoBalance introduces additional memory and computational overhead compared to traditional "pre-combine" methods. Promising future directions include developing more memory- and computation-efficient variants and extending the post-combine principle to broader multi-task learning problems beyond PINNs.

**Ethics Statement**: We have read, understood, and fully adhere to the ICLR Code of Ethics. Our study does not involve human participants, the use of sensitive data (e.g., personal, medical, financial, or otherwise confidential information), or the deployment of any systems in real-world environments. We have conducted a proactive ethical assessment of our work and identified no foreseeable harms, privacy risks, or fairness-related issues (e.g., biases in data, methods, or results) within the scope of this study.

**Reproducibility Statement**: We provide the algorithm details in Appendix B, PDEs details in Appendix A.2.1, and implementation details (including training procedure and hyperparameter settings) in Appendix A.4. We will release the full code for reproduction upon acceptance.

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

# A    APPENDIX

## A.1    INVERSE PROBLEM BY PINN

Recall that in Eq. (2), a PINN aims to solve the following PDE:

$$\begin{aligned}\mathcal{D}[u(x);x] &= 0, & x \in \Omega, \\ \mathcal{B}[u(x);x] &= 0, & x \in \partial\Omega.\end{aligned} \tag{5}$$

Here $u(x)$ is unknown and aimed to solve for, where we call this a forward problem.

In contrast, we are also interested in the cases where we have some observed values of $u(x)$, and aim to infer unknown parameters $a$ of the PDE system (e.g., material properties, source terms, or diffusion coefficients), where $a$ can be a constant or a function of $x$. In other words, PINN for an inverse problem is to solve:

$$\begin{aligned}\mathcal{D}[u(x);x,a] &= 0, & x \in \Omega, \\ \mathcal{B}[u(x);x,a] &= 0, & x \in \partial\Omega.\end{aligned} \tag{6}$$

PINN in the inverse problem aims to solve the following least-squares problem:

$$\underset{w \in \mathbb{R}^p}{\text{minimize}} \quad L(w) := \lambda_F L_{\text{res}}(w) + \lambda_{\text{data}} L_{\text{data}}(w), \tag{7}$$

where the residual loss $L_{\text{res}}$ is defined similarly to Eq. (2), with the coefficient $a$ either treated as an additional learnable parameter if it is constant, or represented by a separate neural network if it depends on $x$. Thus, in the inverse problem setting, there are typically two neural networks: one for approximating the solution $u(x)$ and another for estimating the coefficient $a(x)$.

The data loss $L_{\text{data}}$ is defined over a set of observation points $\{x_{\text{data}}^j\}_{j=1}^{N_{\text{data}}} \subset \Omega$:

$$L_{\text{data}}(\theta) := \frac{1}{N_{\text{data}}} \sum_{j=1}^{N_{\text{data}}} \left( u(x_{\text{data}}^j; w) - u_{\text{obs}}(x_{\text{data}}^j) \right)^2. \tag{8}$$

It is worth noting that, in practice, collecting observations of $u(x)$ can be expensive and noisy, so the number of data points in $L_{\text{data}}$ is typically much smaller than that in the boundary loss, i.e., $N_{\text{obs}} \ll N_b$. Furthermore, the observed data are assumed to be corrupted by Gaussian noise.

## A.2    EXPERIMENT DETAILS

### A.2.1    DATASET DETAILS

**1D reaction-diffusion system:** The reaction-diffusion system, a parabolic PDE, models the macroscopic behavior of particles undergoing both Brownian motion and chemical reactions. It finds wide applications across diverse fields, including information theory, materials science, and biophysics. In this section, we will consider the following system, identical to that presented in Yu et al. (2020).

$$\begin{aligned}u_t &= u_{xx} + R(x,t), & x \in [-\pi, \pi],\ t \in [0,1], \tag{9} \\ u(\pi,t) &= u(-\pi,t) = 0, \tag{10}\end{aligned}$$

$$u(x,0) = \sum_{n=1}^{4} \frac{\sin(nx)}{n} + \frac{\sin(8x)}{8}, \tag{11}$$

where $R(x,t)$ represents the reaction term:

$$R(x,t) = e^{-t} \left[ \frac{3}{2} \sin(2x) + \frac{8}{3} \sin(3x) + \frac{15}{4} \sin(4x) + \frac{63}{8} \sin(8x) \right]. \tag{12}$$

The analytical solution to this system is given by:

$$u(x,t) = e^{-t} \left( \sum_{n=1}^{4} \frac{\sin(nx)}{n} + \frac{\sin(8x)}{8} \right). \tag{13}$$

**2D Helmholtz equation:** The Helmholtz equation, a time-independent partial differential equation, describes key aspects of wave propagation and diffusion processes—serving as the time-separated counterpart to time-dependent models of such phenomena. While it itself is defined within a spatial domain, it underpins analyses of systems that would otherwise be framed in combined spatial-temporal domains. We thus focus on the following 2D form of the Helmholtz equation:

$$u_{xx} + u_{yy} + k^2 u - q(x, y) = 0, \quad (x, y) \in \Omega, \tag{14}$$

$$u(x, y) = 0, \quad (x, y) \in \partial\Omega, \tag{15}$$

where

$$q(x, y) = (k^2 - 2(a\pi)^2) \sin(a\pi x) \sin(a\pi y). \tag{16}$$

We set $k = 1$ and $\Omega = [-1, 1] \times [-1, 1]$ and the exact solution to the equation is

$$u(x, y) = \sin(a_1 \pi x) \sin(a_2 \pi y). \tag{17}$$

In this study, we consider $a_1 = a_2 = 5$.

**Poisson inverse problem (PInv):** We study an inverse problem, specifically targeting the reconstruction of spatially varying coefficients in a 2D Poisson equation. We follow the benchmark setup proposed in Zhongkai et al. (2024). The governing PDE is given by:

$$-\nabla(a\nabla u) = f, \quad (x, y) \in \Omega, \tag{18}$$

where $\Omega = [0, 1]^2$, and the solution is prescribed as $u(x, y) = \sin(\pi x) \sin(\pi y)$. The source term $f$ is derived accordingly:

$$f = \frac{2\pi^2 \sin(\pi x) \sin(\pi y)}{1 + x^2 + y^2 + (x-1)^2 + (y-1)^2} \tag{19}$$

$$+ \frac{2\pi\big((2x-1)\cos(\pi x)\sin(\pi y) + (2y-1)\cos(\pi y)\sin(\pi x)\big)}{(1 + x^2 + y^2 + (x-1)^2 + (y-1)^2)^2}. \tag{20}$$

Our objective is to infer the unknown diffusion coefficient $a(x, y)$ from sparse observations of the solution $u(x, y)$ and the known source term $f(x, y)$. The ground truth of this diffusion coefficient is given by:

$$a(x, y) = \frac{1}{1 + x^2 + y^2 + (x-1)^2 + (y-1)^2}. \tag{21}$$

Moreover, as indicated in Zhongkai et al. (2024), enforcing the boundary condition for $a(x, y)$ is essential to ensure the uniqueness of the inverse solution. The boundary condition is prescribed as:

$$a(x, y) = \frac{1}{1 + x^2 + y^2 + (x-1)^2 + (y-1)^2}, \quad (x, y) \in \partial\Omega. \tag{22}$$

### A.3 BASELINE DETAILS

#### A.3.1 LOSS/GRADIENT BALANCE BASELINES

We outline and describe concisely the baseline models of the balancing methods in Table 1:

- **EW**: Assigns equal global weight $\lambda_i = 1$ to all loss functions, where $\lambda_i$ denotes the weight of the $i$-th loss component.
- **NTK-PINN** (Wang et al., 2022): Dynamically adjusts loss weights using Neural Tangent Kernel (NTK) properties to balance the convergence rates of different components during training.
- **DWA** (Liu et al., 2019): Adapts global loss weights based on recent task performance trends, ensuring balanced progress across all loss components.
- **MGDA** (Désidéri, 2012): Aligns multiple gradients by finding a common descent direction, coordinating updates across tasks.

- **PCGrad** (Yu et al., 2020): Projects conflicting gradients onto the normal plane of others to dynamically resolve task conflicts.

- **IMTL-G** (Liu et al., 2021b): Iteratively normalizes and adjusts gradient directions to refine updates and balance contributions from all tasks.

- **ConFIG** (Liu et al., 2025): Uses conflict-free gradient adjustment based on gradient properties (e.g., pseudoinverse, orthogonal components) to optimize both update direction and magnitude, balancing convergence rates across loss components.

### A.3.2 PINN BASELINES

We outline and describe concisely the baseline models of PINN we test in Table 2:

- **RBA-PINN** (Residual-based Attentional PINN) (Anagnostopoulos et al., 2024): Applies adaptive point-wise weights to the loss functions based on residual magnitude.

- **CWP-PINN** (Convolutional-Weighting PINN) (Si & Yan, 2025a): Extends point-wise weighting with convolutional operations and incorporates a resampling strategy.

- **gPINN** (Gradient-Enhanced PINN) (Yu et al., 2020): Integrates gradient information from PDE residuals into the loss, improving predictive accuracy and training dynamics.

- **RoPINN** (Region-Optimized PINN) (Wu et al., 2024): Expands optimization from discrete points to surrounding regions, reducing generalization errors and addressing high-order PDE constraints.

- **CoPINN** (Cognitive PINN) (Duan et al., 2025): Mimics human easy-to-hard learning by evaluating sample difficulty via PDE residual gradients and applying a cognitive training scheduler to enhance PDE-solving performance.

- **DF-PINN** (Deep Fuzzy PINN) (Wu et al., 2025): Introduces fuzzy neural networks for PDE solving, effectively managing uncertainty in simulation-generated data.

### A.4 IMPLEMENTATION DETAILS

**Training and testing points.** For 2D spatial or 1D spatio-temporal PDEs, test points are generated on a uniform $300 \times 300$ grid (90,000 points). For higher-dimensional problems, 90,000 test points are randomly sampled to balance computational and memory requirements. Training points are also randomly selected, with $N_f$, $N_b$, and $N_0$ denoting the numbers of residual, boundary, and initial condition points, respectively. Note that the training points are not necessarily included in the testing set.

For each PDE, every model is trained and evaluated across three independent trials using different random seeds, and the best-performing result among the trials is reported. Performance is measured using the Mean Squared Error (MSE) and the $L^\infty$ norm, defined as

$$\text{MSE} = \sum_{k=1}^{N} |\hat{u}(\mathbf{x}_k, t_k) - u(\mathbf{x}_k, t_k)|^2, \tag{23}$$

$$L^\infty \text{ norm} = \max_{k=1,2,\ldots,N} |\hat{u}(\mathbf{x}_k, t_k) - u(\mathbf{x}_k, t_k)|, \tag{24}$$

where $u$ denotes the ground truth solution, $\hat{u}$ is the prediction from the model, and $N$ is the number of testing points. All numerical experiments were performed on a computing platform equipped with 2 NVIDIA RTX 4080 GPUs.

To ensure a fair comparison, all models for each PDE are trained using the same hyperparameter settings.

- **1D reaction-diffusion system**: We use a 3-layer network with 50 neurons per hidden layer, with $N_f = 2,000$ and $N_b = 100$. The loss weights are $\lambda_F = 5$ and $\lambda_B = 1$. The learning rate is warmed up from $1 \times 10^{-4}$ to $1 \times 10^{-2}$ over 1,500 iterations, then decayed exponentially by 0.75 every 1,000 iterations (triggered every 50), with a minimum of $5 \times 10^{-5}$. All models are trained for 30k iterations.

- **2D Helmholtz equation**: We use a 3-layer network with 50 neurons per hidden layer, with $N_f = 2,000$ and $N_b = 400$. The loss weights are $\lambda_F = 1$ and $\lambda_B = 1$. The learning rate is warmed up from $1 \times 10^{-4}$ to $1 \times 10^{-2}$ over 1,500 iterations, then decayed exponentially by 0.75 every 1,000 iterations (triggered every 50), with a minimum of $1 \times 10^{-5}$. All models are trained for 30k iterations.

- **2D Poisson inverse problem**: We use a 4-layer network with 50 neurons per hidden layer to reconstruct $a(x, y)$, following the data setup in Si & Yan (2025a). Specifically, $N_f = 100$ and $N_{\text{data}} = 70$ (60 interior data for $u(x, y)$ and 10 boundary data for $a(x, y)$). We add a Gaussian noise with variance 0.01 to the observed $u(x, y)$. The loss weights are $\lambda_F = 1$ and $\lambda_{\text{data}} = 10$. The learning rate is warmed up from $1 \times 10^{-4}$ to $1 \times 10^{-2}$ over 1,500 iterations, then decayed exponentially by 0.75 every 1,000 iterations (triggered every 50), with a floor of $5 \times 10^{-5}$. All models are trained for 30k iterations.

### A.4.1 VISUALIZATION OF AUTO-ADAMW AND BALANCE BASELINE

We select one representative baseline from each category of balancing methods for visualization. Specifically, we compare DWA (loss-balancing) and IMTL-G (gradient-balancing) with our Auto-AdamW to illustrate performance differences.

**1D Reaction-diffusion system:**

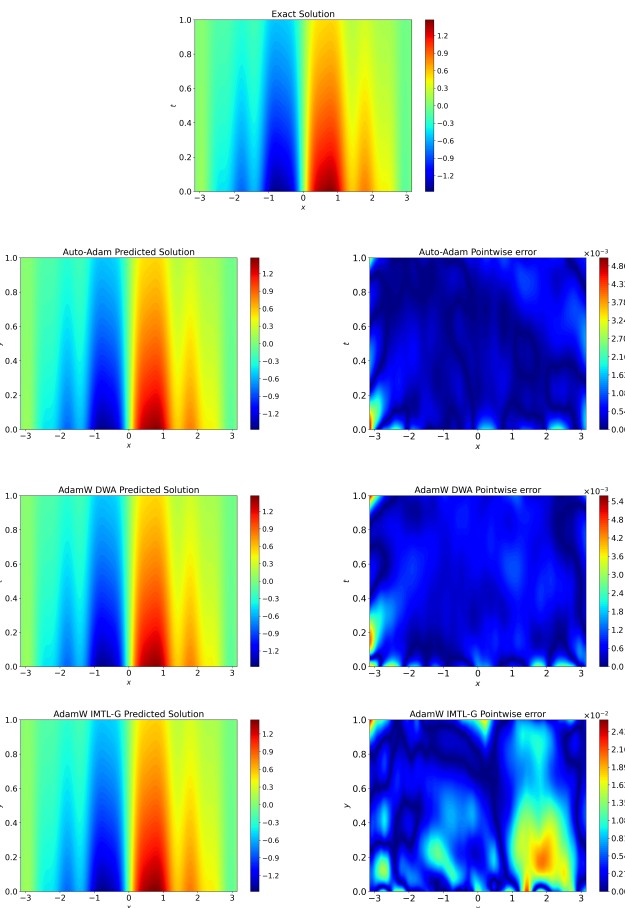

Figure 4: Heatmap of the 1D reaction-diffusion system. **Top Row**: Exact solution. Other Rows: Predicted solutions by Auto-AdamW (**Second Row**), DWA (**Third Row**), and IMTL-G (**Bottom Row**), along with their corresponding point-wise errors.

**2D Helmholtz equation:**

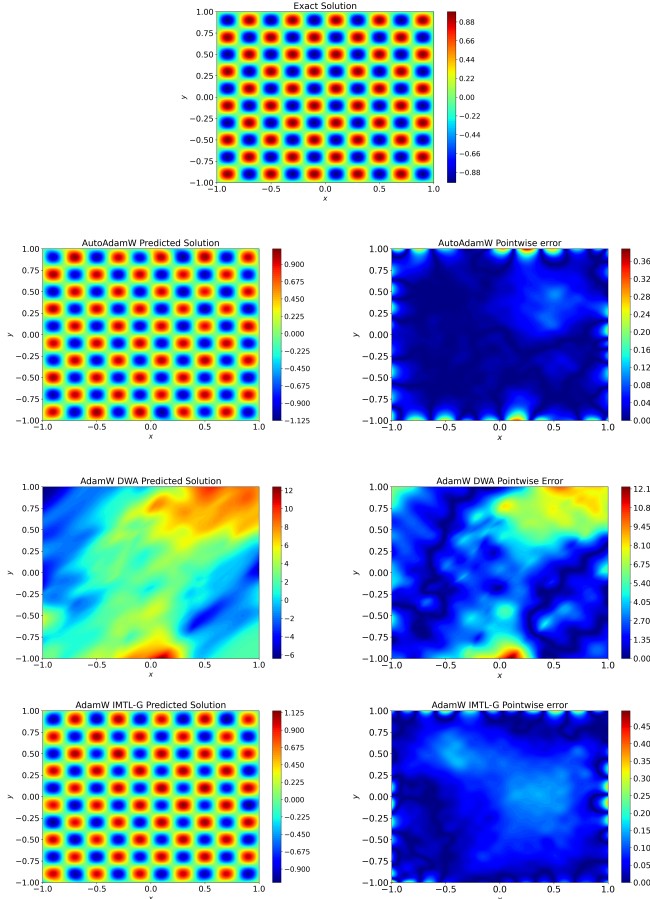

Figure 5: Heatmap of the 2D Helmholtz equation. **Top Row**: Exact solution. Other Rows: Predicted solutions by Auto-AdamW (**Second Row**), DWA (**Third Row**), and IMTL-G (**Bottom Row**), along with their corresponding point-wise errors.

**2D Poisson inverse problem:**

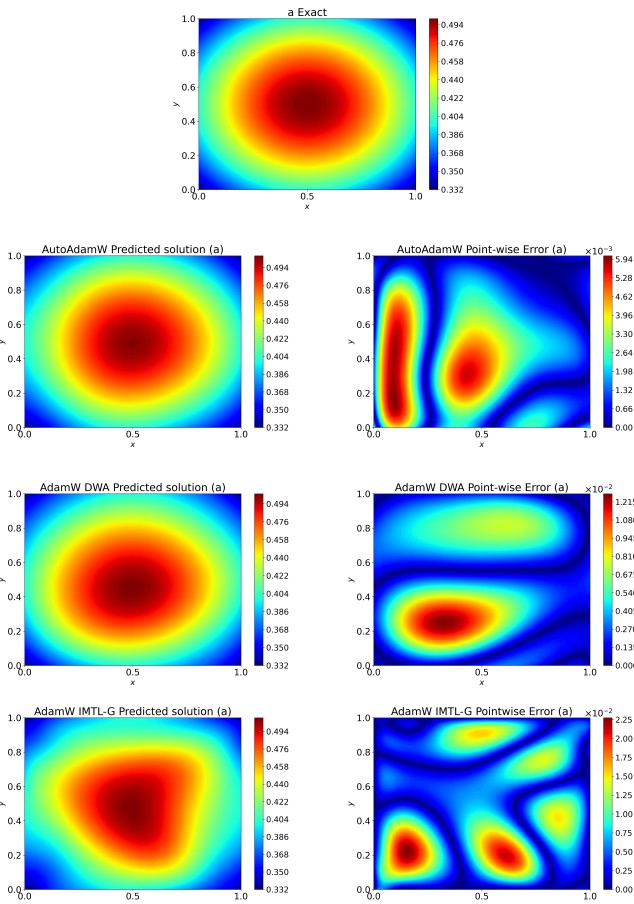

Figure 6: Qualitative comparison of reconstructed diffusion coefficients for the 2D Poisson inverse problem. **Top Row:** Ground-truth field. Other Rows: Reconstructions by AutoAdamW (**Second Row**), DWA (**Third Row**), and IMTL-G (**Bottom Row**) with corresponding point-wise errors, showing progressively decreasing accuracy.

### A.4.2    VISUALIZATION OF AUTO-ADAMW WITH THE PINN BASELINE

In this section, we present heatmaps illustrating the performance of PINN baselines using traditional AdamW and Auto-AdamW. To provide a clear and representative comparison, we select one typical example from the adaptive weighting category, RBA-PINN, and one from the novel loss function category, RoPINN.

- **1D reaction-diffusion system:**

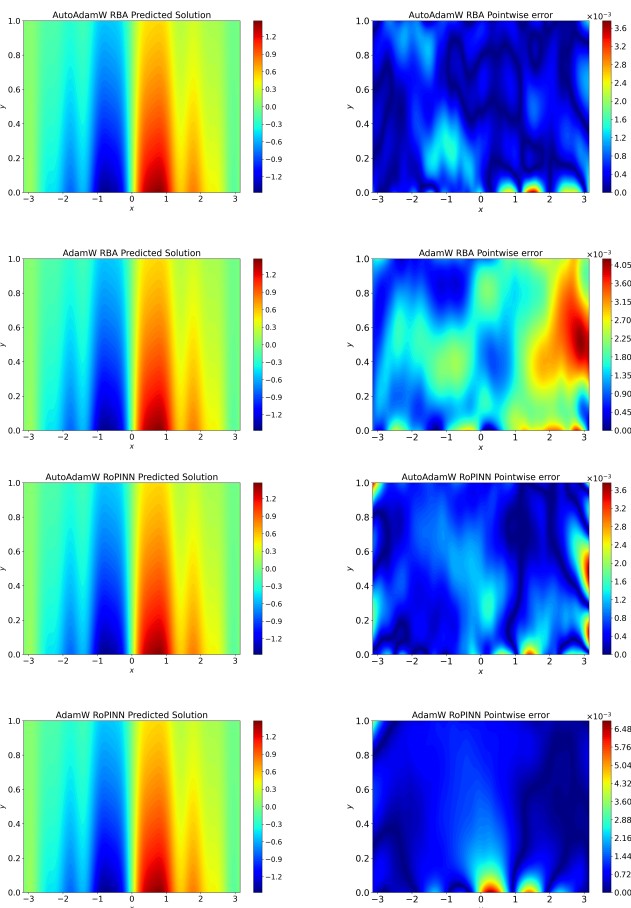

Figure 7: Heatmap of the 1D reaction-diffusion system. **Top Two Rows**: RBA-PINN predictions using Auto-AdamW and traditional AdamW, with corresponding point-wise errors. **Bottom Two Rows**: RoPINN predictions using Auto-AdamW and traditional AdamW, with corresponding point-wise errors.

- **2D Helmholtz equation:**

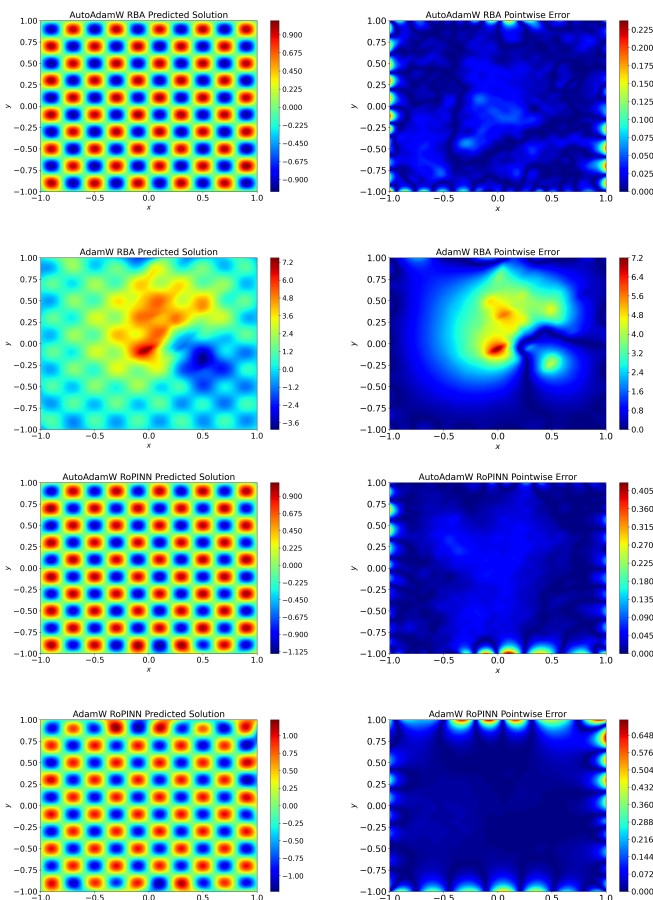

Figure 8: Heatmap of the 2D Helmholtz equation. **Top Two Rows**: RBA-PINN predictions using Auto-AdamW and traditional AdamW, with corresponding point-wise errors. **Bottom Two Rows**: RoPINN predictions using Auto-AdamW and traditional AdamW, with corresponding point-wise errors.

• **2D Poisson inverse problem:**

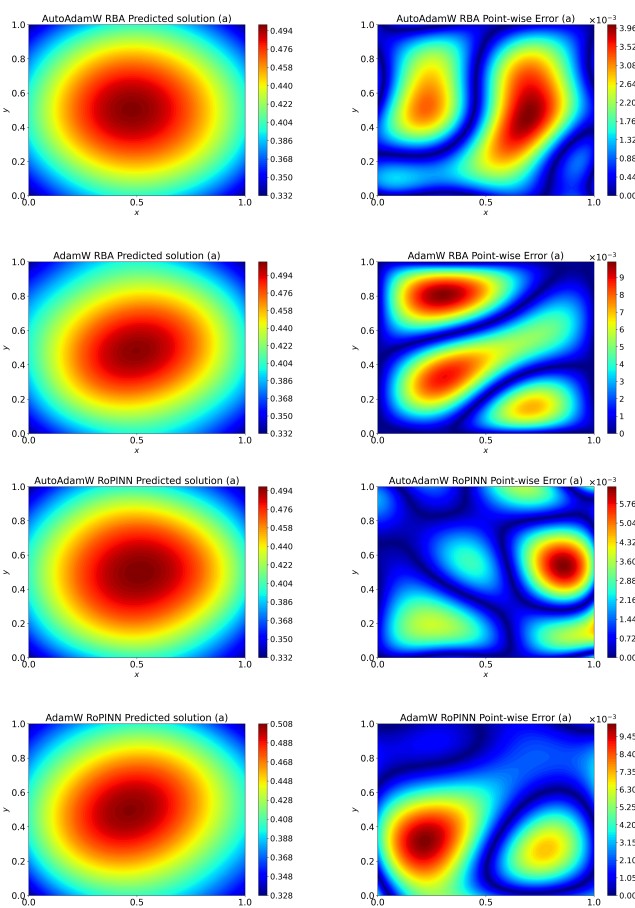

Figure 9: Heatmap of the 2D Poisson inverse problem. **Top Two Rows**: RBA-PINN predictions using Auto-AdamW and traditional AdamW, with corresponding point-wise errors. **Bottom Two Rows**: RoPINN predictions using Auto-AdamW and traditional AdamW, with corresponding point-wise errors.

# B ALGORITHM AND PROOFS

## B.1 ALGORITHM

---
**Algorithm 1** AutoAdamW for $n$ loss terms
---
1: **Input:** initialization $\boldsymbol{w}^0$, number of iterations $T$, $\beta_1$, $\beta_2$, $\epsilon$, $\gamma$, $n$, $\{\eta^k\}_{k=1}^T$
2: Initialize $\boldsymbol{m}_i^0, \boldsymbol{v}_i^0 \leftarrow \boldsymbol{0}$ for $i = 1, \ldots, n$
3: **for** $k = 1$ **to** $T$ **do**
4:     **for** $i = 1$ **to** $n$ **do**
5:        $\boldsymbol{m}_i^k = \beta_1 \boldsymbol{m}_i^{k-1} + (1 - \beta_1)\nabla\mathcal{L}_i(\boldsymbol{w}^{k-1})$
6:        $\boldsymbol{v}_i^k = \beta_2 \boldsymbol{v}_i^{k-1} + (1 - \beta_2)\nabla\mathcal{L}_i(\boldsymbol{w}^{k-1}) \odot \nabla\mathcal{L}_i(\boldsymbol{w}^{k-1})$
7:        $\hat{\boldsymbol{m}}_i^k = \frac{\boldsymbol{m}_i^k}{1-\beta_1^k}, \hat{\boldsymbol{v}}_i^k = \frac{\boldsymbol{v}_i^k}{1-\beta_2^k}$
8:        $d_i^k = \frac{\hat{\boldsymbol{m}}_i^k}{\sqrt{\hat{\boldsymbol{v}}_i^k+\epsilon}}$
9:     **end for**
10:    $\tilde{\boldsymbol{d}}^k = \frac{1}{n}\sum_{i=1}^n d_i^k$
11:    $\boldsymbol{w}^k = (1 - \eta^k\gamma)\boldsymbol{w}^{k-1} - \eta^k\tilde{\boldsymbol{d}}^k$
12: **end for**
---

---
**Algorithm 2** Adam without bias correction
---
1: **Input:** initialization $\boldsymbol{w}^0$, number of iterations $T$, $\beta_1$, $\beta_2$, $n$, $\eta$
2: Initialize $\boldsymbol{m}^0 \leftarrow \nabla\mathcal{L}(\boldsymbol{w}^0), \boldsymbol{v}^0 \leftarrow \nabla\mathcal{L}(\boldsymbol{w}^0) \odot \nabla\mathcal{L}(\boldsymbol{w}^0)$
3: **for** $k = 1$ **to** $T$ **do**
4:    $\boldsymbol{m}^k = \beta_1 \boldsymbol{m}^{k-1} + (1 - \beta_1)\nabla\mathcal{L}(\boldsymbol{w}^{k-1})$
5:    $\boldsymbol{v}^k = \beta_2 \boldsymbol{v}^{k-1} + (1 - \beta_2)\nabla\mathcal{L}(\boldsymbol{w}^{k-1}) \odot \nabla\mathcal{L}(\boldsymbol{w}^{k-1})$
6:    $\boldsymbol{w}^k = \boldsymbol{w}^{k-1} - \eta\frac{\boldsymbol{m}^k}{\sqrt{\boldsymbol{v}^k}}$
7: **end for**
---

---
**Algorithm 3** AutoAdam for $n$ loss functions without bias correction
---
1: **Input:** initialization $\boldsymbol{w}^0$, number of iterations $T$, $\beta_1$, $\beta_2$, $n$, $\eta$
2: Initialize $\boldsymbol{m}_i^0 \leftarrow \nabla\mathcal{L}_i(\boldsymbol{w}^0), \boldsymbol{v}_i^0 \leftarrow \nabla\mathcal{L}_i(\boldsymbol{w}^0) \odot \nabla\mathcal{L}_i(\boldsymbol{w}^0)$
3: **for** $k = 1$ **to** $T$ **do**
4:     **for** $i = 1$ **to** $n$ **do**
5:        $\boldsymbol{m}_i^k = \beta_1 \boldsymbol{m}_i^{k-1} + (1 - \beta_1)\nabla\mathcal{L}_i(\boldsymbol{w}^{k-1})$
6:        $\boldsymbol{v}_i^k = \beta_2 \boldsymbol{v}_i^{k-1} + (1 - \beta_2)\nabla\mathcal{L}_i(\boldsymbol{w}^{k-1}) \odot \nabla\mathcal{L}_i(\boldsymbol{w}^{k-1})$
7:        $\boldsymbol{d}_i^k = \frac{\boldsymbol{m}_i^k}{\sqrt{\boldsymbol{v}_i^k}}$
8:     **end for**
9:    $\tilde{\boldsymbol{d}}^k = \frac{1}{n}\sum_{i=1}^n d_i^k$
10:   $\boldsymbol{w}^k = \boldsymbol{w}^{k-1} - \eta\tilde{\boldsymbol{d}}^k$
11: **end for**
---

## B.2 PROOF OF THEOREM 1

*Proof.* We analyze the one-step convergence rate of both algorithms on a quadratic objective, assuming the minimizer is $\boldsymbol{w}^* = \boldsymbol{0}$.

**AutoAdam:** Under the specified conditions ($\beta_1 = 0, \beta_2 = 1$), the update rule for $\boldsymbol{w}^t$ in Algorithm 3 reduces to a linear iteration:

$$\boldsymbol{w}^{t+1} = \boldsymbol{w}^t - \eta\left((D_1^0)^{-1}\boldsymbol{w}^t + (D_2^0)^{-1}A^T A\boldsymbol{w}^t\right) = (I - \eta M_{\text{AutoAdam}})\boldsymbol{w}^t,$$

where $D_1^0 = \text{diag}(|\nabla L_1(\boldsymbol{w}^0)|)$, $D_2^0 = \text{diag}(|\nabla L_2(\boldsymbol{w}^0)|)$, and the effective preconditioned Hessian is,

$$M_{\text{AutoAdam}} = (D_1^0)^{-1} + (D_2^0)^{-1}A^T A = (D_2^0)^{-1}\left(D_2^0(D_1^0)^{-1} + A^T A\right).$$

Since $M_{\text{AutoAdam}}$ is not symmetric, we introduce the symmetric matrix

$$W = \left(D_2^0(D_1^0)^{-1} + A^T A\right)$$

and define the corresponding weighted norm $\|\boldsymbol{w}\|_W^2 = \boldsymbol{w}^T W \boldsymbol{w}$ to measure convergence.

We then examine the contraction of the weighted distance to the minimizer $\boldsymbol{w}^* = \mathbf{0}$. The ratio at each step is

$$
\begin{aligned}
\frac{\|\boldsymbol{w}^{t+1}\|_W}{\|\boldsymbol{w}^t\|_W} &= \frac{\|(I - \eta(D_2^0)^{-1}W))\boldsymbol{w}^t\|_W}{\|\boldsymbol{w}^t\|_W} \\
&= \frac{\|(I - \eta\sqrt{W}(D_2^0)^{-1}\sqrt{W}))\sqrt{W}\boldsymbol{w}^t\|_2}{\|\sqrt{W}\boldsymbol{w}^t\|_2} \\
&\leq \max(|1 - \eta\lambda_{\min}(\sqrt{W}(D_2^0)^{-1}\sqrt{W})|, |1 - \eta\lambda_{\max}(\sqrt{W}(D_2^0)^{-1}\sqrt{W})|).
\end{aligned}
\tag{25}
$$

To achieve the fastest guaranteed convergence, we choose the step size $\eta^*$ that minimizes the upper bound. Since the bound is quadratic in $\eta$, the minimizer is

$$\eta^* = \frac{2}{\lambda_{\max}(\sqrt{W}(D_2^0)^{-1}\sqrt{W}) + \lambda_{\min}(\sqrt{W}(D_2^0)^{-1}\sqrt{W})}.$$

Substituting $\eta^*$ back into the inequality gives the following one-step convergence guarantee:

$$\frac{\|\boldsymbol{w}^{t+1}\|_W}{\|\boldsymbol{w}^t\|_W} \leq \frac{\lambda_{\max}(\sqrt{W}(D_2^0)^{-1}\sqrt{W}) - \lambda_{\min}(\sqrt{W}(D_2^0)^{-1}\sqrt{W})}{\lambda_{\max}(\sqrt{W}(D_2^0)^{-1}\sqrt{W}) + \lambda_{\min}(\sqrt{W}(D_2^0)^{-1}\sqrt{W})}.
\tag{26}$$

Next, observe that

$$
\begin{aligned}
\lambda_{\max}(\sqrt{W}(D_2^0)^{-1}\sqrt{W}) &= \lambda_{\max}((D_2^0)^{-1/2}\sqrt{W}\sqrt{W}(D_2^0)^{-1/2}) \\
&= \lambda_{\max}((D_1^0)^{-1} + (D_2^0)^{-1/2}A^T A(D_2^0)^{-1/2}),
\end{aligned}
$$

and similarly,

$$\lambda_{\min}(\sqrt{W}(D_2^0)^{-1}\sqrt{W}) = \lambda_{\min}((D_1^0)^{-1} + (D_2^0)^{-1/2}A^T A(D_2^0)^{-1/2}).$$

Substituting these expressions into the one-step convergence bound gives

$$\frac{\|\boldsymbol{w}^{t+1}\|_W}{\|\boldsymbol{w}^t\|_W} \leq \frac{\kappa((D_1^0)^{-1} + (D_2^0)^{-1/2}A^T A(D_2^0)^{-1/2}) - 1}{\kappa((D_1^0)^{-1} + (D_2^0)^{-1/2}A^T A(D_2^0)^{-1/2}) + 1},
\tag{27}$$

where $\kappa(\cdot)$ denotes the condition number.

**Adam:** A similar analysis applies to the Adam optimizer (Algorithm 2) under the same assumptions. Its update rule can be written as a linear iteration:

$$\boldsymbol{w}^{t+1} = (I - \eta M_{\text{Adam}})\boldsymbol{w}^t,$$

where $D^0 = \text{diag}(|\nabla L(\boldsymbol{w}^0)|)$ and the effective preconditioned Hessian is,

$$M_{\text{Adam}} = (D^0)^{-1}(I + A^T A).$$

Following the same derivation as for AutoAdam, we define the symmetrized matrix

$$W_{\text{Adam}} = I + A^T A,$$

and measure convergence in the weighted norm $\|\boldsymbol{w}\|_{W_{\text{Adam}}}^2 = \boldsymbol{w}^\top W_{\text{Adam}}\boldsymbol{w}$. Then the one-step convergence satisfies

$$\frac{\|\boldsymbol{w}^{t+1}\|_{W_{\text{Adam}}}}{\|\boldsymbol{w}^t\|_{W_{\text{Adam}}}} \leq \frac{\kappa((D^0)^{-1} + (D^0)^{-1/2}A^T A(D^0)^{-1/2}) - 1}{\kappa((D^0)^{-1} + (D^0)^{-1/2}A^T A(D^0)^{-1/2}) + 1}.
\tag{28}$$

$\square$

### B.3   PROOF OF COROLLARY 1.1

*Proof.* We provide a lower bound on the one-step convergence rate, which is governed by the effective conditioning of the preconditioned Hessian. We analyze AutoAdam and Adam separately under their respective Bounded Initialization assumptions.

**AutoAdam:** The Bounded Initialization assumption for AutoAdam states that there exist constants $C_{1,1}, C_{1,2}, C_{2,1}, C_{2,2} > 0$ such that

$$C_{1,1}I \preceq D_1^0 \preceq C_{1,2}I,$$
$$C_{2,1}\lambda_{\max}(A^TA)I \preceq D_2^0 \preceq C_{2,2}\lambda_{\max}(A^TA)I.$$

Under this assumption, we can bound the condition number of the effective preconditioned Hessian:

$$\lambda_{\max}((D_1^0)^{-1} + (D_2^0)^{-1/2}A^TA(D_2^0)^{-1/2}) \leq \frac{1}{C_{1,1}} + \frac{\lambda_{\max}(A^TA)}{C_{2,1}\lambda_{\max}(A^TA)} = \frac{1}{C_{1,1}} + \frac{1}{C_{2,1}},$$
$$\lambda_{\min}((D_1^0)^{-1} + (D_2^0)^{-1/2}A^TA(D_2^0)^{-1/2}) \geq \frac{1}{C_{1,2}}.$$

Thus, the condition number satisfies

$$\kappa((D_1^0)^{-1} + (D_2^0)^{-1/2}A^TA(D_2^0)^{-1/2}) \leq \frac{C_{1,2}}{C_{1,1}} + \frac{C_{1,2}}{C_{2,1}}.$$

**Adam:** The Bounded Initialization assumption for Adam states that there exist constants $C_1, C_2 > 0$ such that
$$C_1\lambda_{\max}(I + A^TA)I \preceq D^0 \preceq C_2\lambda_{\max}(I + A^TA)I.$$

Under this assumption, we can similarly bound the condition number:

$$\lambda_{\max}((D^0)^{-1} + (D^0)^{-1/2}A^TA(D^0)^{-1/2}) \leq \frac{\lambda_{\max}(I + A^TA)}{C_1\lambda_{\max}(I + A^TA)} = \frac{1}{C_1},$$
$$\lambda_{\min}((D^0)^{-1} + (D^0)^{-1/2}A^TA(D^0)^{-1/2}) \geq \frac{\lambda_{\min}(I + A^TA)}{C_2\lambda_{\max}(I + A^TA)}.$$

Hence, the condition number is bounded by

$$\kappa((D^0)^{-1} + (D^0)^{-1/2}A^TA(D^0)^{-1/2}) \leq \frac{C_2}{C_1}\kappa(I + A^TA).$$

$\square$

## C   USE OF LLMs

We acknowledge that we used LLMs solely for polishing purposes, such as refining grammar, phrasing, and clarity. All core ideas, analytical work, and substantive content were developed and written entirely by the authors.

