# OpenReview forum: "AutoBalance: An Automatic Balancing Framework for Training Physics-Informed Neural Networks"
_ICLR.cc/2026/Conference — ICLR 2026 Conference Withdrawn Submission_

### Official Review · Reviewer_8LYs · 2025-10-15

**Soundness:** 3
**Presentation:** 3
**Contribution:** 2
**Rating:** 2
**Confidence:** 4

**Summary:**

This paper contends that the prevalent pre-combine strategy adopted in conventional PINN frameworks, wherein the gradients of the PDE residual loss and boundary condition loss are manipulated prior to their input into optimizers such as Adam, is inherently suboptimal. To remedy this limitation, the authors introduce AutoBalance, a post-combine optimization scheme that independently applies the optimizer to the residual and boundary gradients before aggregating the optimized updates. Empirical evaluations demonstrate that AutoBalance consistently surpasses pre-combined baselines and yields further performance gains when integrated with various PINN variants.

**Strengths:**

**Strength**

**S1.** The authors provided a theoretical analysis of the advantages of the proposed post-combine strategy under a simplified quadratic objective. This analysis offers a clear understanding of the motivation behind AutoBalance, and the theoretical insights are further supported by empirical evidence showing that AutoBalance achieves a smaller condition number compared to the baseline (Adam).

**S2.** In the presented experiments, AutoBalance not only outperforms existing loss-balancing and gradient-manipulation methods but also demonstrates enhanced performance when combined with various PINN variants.

**S3.** Beyond PINNs, AutoBalance can be readily extended to other learning paradigms involving multiple losses, such as multi-task learning.

**Weaknesses:**

**Weakness**

**W1.** MultiAdam [1] essentially applies a post-combine strategy to PINNs and is algorithmically equivalent to AutoBalance (more exactly, same with AutoAdam). However, the paper fails to acknowledge this prior work, indicating a lack of comprehensive review of related studies. While MultiAdam does not include analyses such as Hessian spectral characterization, and thus the novelty of the present work remains justifiable, a clear discussion of similar approaches and a more thorough rewriting of the related work section are necessary for completeness.

**W2.** The set of loss balancing and gradient manipulation methods compared in the experiments is outdated and omits several recent and relevant approaches. For instance, PINNACLE [2] and ReLoBRaLo [3] represent state-of-the-art loss balancing methods, while Aligned-MTL [4] and DCGD [5] are strong gradient manipulation baselines. Including these methods in the comparison would significantly strengthen the empirical evaluation.

**W3.** The benchmark equations used in the paper are limited to three relatively simple cases. A more rigorous evaluation on high-dimensional or particularly challenging equations, where vanilla PINNs (EW in this paper) typically fail, would provide stronger evidence of AutoBalance’s robustness. For example, employing benchmark suites such as those presented in [5] would more convincingly demonstrate the superiority of AutoBalance across diverse and difficult PDE settings.

**W4**. The reported results are based on a single run. Evaluating the performance across multiple random seeds and reporting the corresponding mean and standard deviation would provide a more rigorous and statistically reliable assessment of the method’s effectiveness.

[1] Yao, Jiachen, et al. "Multiadam: Parameter-wise scale-invariant optimizer for multiscale training of physics-informed neural networks." International Conference on Machine Learning. PMLR, 2023.

[2] Lau, Gregory Kang Ruey, et al. "PINNACLE: PINN Adaptive ColLocation and Experimental points selection." The Twelfth International Conference on Learning Representations.

[3] Bischof, Rafael, and Michael A. Kraus. "Multi-objective loss balancing for physics-informed deep learning." Computer Methods in Applied Mechanics and Engineering 439 (2025): 117914.

[4] Senushkin, Dmitry, et al. "Independent component alignment for multi-task learning." Proceedings of the IEEE/CVF Conference on Computer Vision and Pattern Recognition. 2023.

[5] Zhongkai, Hao, et al. "Pinnacle: A comprehensive benchmark of physics-informed neural networks for solving pdes." Advances in Neural Information Processing Systems 37 (2024): 76721-76774.

**Questions:**

**Questions**

**Q1.** In the benchmark presented in [5], which was mentioned in W3, MultiAdam did not outperform other competing methods. Does this imply that the effectiveness of AutoBalance is particularly pronounced when combined with AdamW? If the experiments in Table 1 were conducted using Adam instead of AdamW, would AutoBalance still outperform other methods?

**Q2.** The paper introduces ConFIG as a competitor, but the authors of ConFIG also proposed a momentum-based variant, M-ConFIG, which applies momentum to each gradient before aggregation. Would this approach be considered a pre-combined or post-combined strategy, or neither? Furthermore, M-ConFIG was not included in the performance comparison. Could the authors provide results or insights on how M-ConFIG performs relative to AutoBalance?

**Q3.** Unlike pre-combined strategies, does the post-combined strategy inherently nullify the effects of gradient manipulation techniques? Is it possible to integrate AutoBalance with pre-combined methods such as PCGrad, ConFIG, or MGDA, and if so, what would be the conceptual or practical implications of such integration?

---

### Official Review · Reviewer_jgYo · 2025-10-28

**Soundness:** 2
**Presentation:** 3
**Contribution:** 2
**Rating:** 4
**Confidence:** 4

**Summary:**

This paper targets the long-standing optimization difficulty in training Physics-Informed Neural Networks (PINNs), where multiple loss components (e.g., PDE residuals and boundary conditions) often have heterogeneous curvature, leading to ill-conditioned optimization. The authors argue that the prevailing “pre-combine” paradigm—manipulating gradients before passing them into a single optimizer—disrupts preconditioning. They propose AutoBalance, a “post-combine” training framework that assigns an independent adaptive optimizer to each loss term and aggregates the preconditioned updates. The paper provides theoretical motivation via convergence analysis on quadratic problems, and empirical verification of Hessian spectrum behavior.

**Strengths:**

1. The paper correctly identifies a crucial and under-explored source of PINN training instability—heterogeneous curvature among multiple loss components. This is supported by both theoretical arguments and empirical evidence on Hessian spectra (Fig. 1, Sec. 3.2).

2. Instead of designing complex weighting or gradient projection strategies, the proposed “post-combine” framework leverages existing adaptive optimizers independently for each loss component. This is elegant, orthogonal to architectural or loss design changes, and easy to implement.

3. The authors provide a convergence analysis on quadratic objectives (Theorem 1, Corollary 1.1), showing that AutoBalance achieves better conditioning and faster convergence when curvature heterogeneity is present. This connects the method to a solid mathematical basis, which is often missing in PINN optimization work.

4. The method is shown to be orthogonal to other PINN variants, improving performance when stacked on top of existing models (RBA-PINN, CoPINN, DF-PINN, etc.).

**Weaknesses:**

1. The AutoAdam algorithm is extremely close to existing variants such as MultiAdam or simple multi-optimizer aggregation. The paper should explicitly clarify what differentiates AutoBalance from MultiAdam or other multi-optimizer strategies—especially in terms of algorithmic novelty and theoretical contribution.

2. The benchmarks used (1D reaction-diffusion, 2D Helmholtz, 2D Poisson inverse problem) are still relatively standard and low dimensional. It would strengthen the paper to include higher-dimensional or more complex PDEs (e.g., Navier–Stokes, nonlinear elasticity) to demonstrate scalability and robustness under harsher curvature heterogeneity.

3. The impact of $\beta_2$ (second moment estimate in Adam) is not analyzed, although the method critically depends on preconditioning behavior. It would be useful to show how sensitive AutoBalance is to optimizer hyperparameters and whether the emergent balancing behavior is robust to these variations.

4. While the quadratic analysis is elegant, it does not fully cover the interaction between curvature imbalance and conflicting minimizers, which often occurs in practical PINNs. The paper acknowledges this in passing, but a more comprehensive discussion would make the theoretical argument stronger.

5. Although mentioned briefly in the conclusion, the paper does not provide concrete measurements of memory and computation overhead. Since AutoBalance requires multiple optimizers, this could be non-trivial for large models.

**Questions:**

1. How exactly does AutoBalance differ from MultiAdam beyond implementation detail? Is there a theoretical insight that is novel?

2. How does the method scale to PDE systems with 3D geometry, time-dependence, or coupled equations (e.g., Navier–Stokes)?

3. Have the authors considered other adaptive preconditioners (e.g., Shampoo, AdaHessian)? Would AutoBalance still offer benefits?

4. How sensitive is the auto-balancing behavior to optimizer hyperparameters like $\beta_1$, $\beta_2$, and learning rate?

---

### Official Review · Reviewer_iM8N · 2025-10-31

**Soundness:** 2
**Presentation:** 3
**Contribution:** 2
**Rating:** 4
**Confidence:** 4

**Summary:**

This paper tackles a fundamental issue in the optimization of Physics-Informed Neural Networks (PINNs): the imbalance between heterogeneous loss terms (e.g., PDE residual and boundary condition losses) with distinct curvature spectra. To address this, the paper introduces AutoBalance, a post-combine framework in which each loss term is assigned an independent adaptive optimizer (e.g., AdamW). The individually preconditioned updates are then aggregated, forming a unified update direction that achieves an emergent auto-balancing behavior without introducing extra hyperparameters.

**Strengths:**

S1. **conceptual clarity**: The paper offers a clear conceptual reframing of the loss imbalance problem in PINNs. Identifying “spectral heterogeneity” in the Hessians of multi-objective losses as the core difficulty is insightful. The “pre-combine vs. post-combine” distinction provides an intuitive optimization perspective.

S2. **emergent auto-balancing insight**: I think the observation in Section 3.3 is novel and offers a new insight into the optimization dynamics of PINNs. This phenomenon provides meaningful understanding of why the proposed method achieves stable convergence.

S3. **strong empirical results and orthogonality**: The proposed method achieves consistent improvements over baselines. Furthermore, this paper highlights its orthogonality by showing that AutoBalance enhances the performance of advanced PINN architectures such as RBA-PINN and gPINN.

S4. **simple and easy implementation**: The algorithm is simple to implement and does not rely on additional hyperparameters or tuning.

**Weaknesses:**

W1. **Insufficient literature review**: The paper overlooks several key prior studies that share the same fundamental problem. For example, see [1,2]. Beyond these, there are other closely related works that analyze or propose solutions to address similar issues in PINNs.

[1] Hwang, Youngsik, and Dongyoung Lim. "Dual cone gradient descent for training physics-informed neural networks." Advances in Neural Information Processing Systems 37 (2024): 98563-98595.

[2] Yao, Jiachen, et al. "Multiadam: Parameter-wise scale-invariant optimizer for multiscale training of physics-informed neural networks." International Conference on Machine Learning. PMLR, 2023.

W2. **Limited novelty**: Although the paper frames AutoBalance as a new “post-combine” optimization paradigm,its algorithmic mechanism appears nearly identical to that of MultiAdam [2]. The distinction between the two methods is unclear.

W3. **Theoretical Limitations**: While the simplified quadratic analysis in Section 3.2 offers some intuitive understanding, it falls short of generalizing to the highly non-convex loss landscape of practical PINNs. Consequently, the theoretical justification for why the proposed algorithm influence PINN training dynamics remains quite limited.

In particular, the use of the condition number as a central analytical tool is problematic since  it represents a globla property of the loss surface. Relying on such a global metric to analyze the behavior of adaptive preconditioned optimizer is fundamentally inappropriate. As a result, the extension of the current analysis to the non-convex setting is theoretically weak.

W4. **benchmark scope**: The evaluation is restricted to classical 1D and 2D PDEs. Recent PINN research increasingly emphasizes stiff, high-dimensional domain problems. Demonstrating AutoBalance’s robustness on these more challenging cases would considerably strengthen the contribution and practical relevance.

W5. **Unfair choice of base optimizer across baselines**: All baseline approaches employ Adam as their undlerying optimizer, whereas AutoBalance adopts AdamW as the base optimizer. This discrepancy raises a fairness issue in the comparative evaluation.

**Questions:**

Q1. How does AutoBalance differ from MultiAdam?

Q2. For a fair comparison and to isolate the effect of AdamW, please include additional experiments using Adam (without weight decay) as the base optimizer.

Q3. Could you report the mean ± standard deviation of performance across multiple random seeds, rather than only presenting the best-performing results?

Q4. How is the “AutoAdam Preconditioned Hessian” formally defined? The definition provided in the Proof of Theorem 1 appears to be restricted to a quadratic toy example.

I am open to raising my overall evaluation if the concerns outlined in the Weaknesses and Questions sections stem from a misunderstanding or are adequately addressed in the rebuttal.

---

### Note · Authors · 2026-01-20

I have read and agree with the venue's withdrawal policy on behalf of myself and my co-authors.